# Parametric Entropic Locally Linear Embedding for Small Sample Size Classification

## Abstract

Manifold learning algorithms are powerful non-linear dimensionality reduction methods for unsupervised metric learning. The locally linear embedding (LLE) method uses the local geometry of the linear neighborhood spaces to estimate optimal reconstruction weights for each sample. In the present paper, we propose the parametric entropic LLE (PELLE) method, which adopts the relative entropy instead of the pointwise Euclidean metric to build local entropic covariance matrices. This methodological improvement increases the robustness of the method regarding noise and outliers. Moreover, state-of-the-art algorithms such as UMAP require a large number of samples for convergence to good results due to numerical optimization methods (gradient descent). Results considering 25 distinct real-world datasets indicate that the proposed method is capable of generating superior clustering and classification accuracies compared to existing state-of-the-art methods for dimensionality reduction-based metric learning, especially in datasets with a limited number of samples.

## 1 Introduction

Metric learning refers to the building of adaptive distance functions to a dataset prior to a classification step. Manifold learning algorithms are capable of finding more compact and meaningful representations for the observed data while preserving the intrinsic non-Euclidean geometry of the data. The locally linear embedding (LLE) is one of the first algorithms of this class to be employed to classification tasks through non-linear dimensionality reduction (Roweis and Saul, 2000). However, one of the main caveats of the LLE is that such a method is remarkably sensitive to noise and outliers (Wang et al., 2019). Thus, the LLE method performs poorly in data sets that do not lead to smooth manifolds. The existing literature contains many LLE extensions proposed to overcome such a limitation.

Hessian eigenmaps replace the local covariance matrix used in the computation of the optimal reconstruction weights by the Hessian matrix (i.e. matrix of the second order derivatives), encoding curvature information (Donoho and Grimes, 2003; Wang, 2012; Xing et al., 2016). The local tangent space alignment (LTSA) method was proposed as a generalization of the LLE method. The main difference between the LTSA and LLE method is that in the former the locally linear patch is built by approximating the tangent space at a particular sample through the application of principal component analysis (PCA) on the neighborhood system. As each neighbor contains its own coordinate representation in the tangent space, we use them as a low dimensional representation of the patch. Therefore, all local representations (i.e. patches) are then aligned to a global one (Zhang and Zha, 2004). In addition, the modified LLE (MLLE) still extends the LLE method by introducing multiple linearly independent local weight vectors for each neighborhood of the k-nearest neighbors (KNN) graph. It has been demonstrated that the local geometry of the MLLE is more stable compared to the LLE method (Zhang and Wang, 2006).

Nonetheless, all of these LLE methodological extensions adopt, at some degree, the Euclidean distance as the similarity measure within a linear patch. Thus, in order to incorporate the relative entropy in the estimation of the optimal reconstruction weights, in the present paper we then propose the parametric entropic LLE (PELLE) method. There are three main contributions of the PELLE method to the literature in comparison with existing methods. Firstly, the pointwise Euclidean distance is replaced by a patch-based information-

theoretic distance (KL-divergence), making the method less sensitive to the presence of noise and outliers. Secondly, the multicollinearity phenomenon in data patches degrades the performance of the LLE algorithm, since it is, among other things, one of the causes of ill-conditioned matrices in the estimation of the optimal reconstruction weights. With the incorporation of the relative entropy (KL-divergence), we can mitigate this problem. Finally, results using several real-world data sets indicate that the PELLE method produces superior clustering and classification accuracies for different supervised classifiers compared to the original LLE as well as state-of-the-art manifold learning algorithms, such as the ISOMAP (Tenenbaum et al., 2000) and UMAP (McInnes et al., 2018).

The remainder of the paper is organized as follows. Section 2 mentions related work, focusing on the derivation of the regular LLE algorithm. Section 3 details the proposed PELLE method algorithm. Section 4 reports computational experiments and related results. Section 5 concludes and suggests future research directions in metric learning-based dimensionality reduction.

## 2 Related work

In this section, we discuss the LLE algorithm, detailing its motivations and mathematical derivation.

### 2.1 Locally linear embedding (LLE) algorithm

The LLE algorithm consists of a local method, in which the new coordinates of any $\vec{x}_i \in R^m$ depends only on the neighborhood of the respective point. The main hypothesis behind the LLE is that of a sufficiently high density of samples, being expected that a vector $\vec{x}_i$ and its neighbors define a linear patch, all belonging to an Euclidean subspace (Roweis and Saul, 2000). Thus, we may characterize the local geometry by linear coefficients, as follows:

$$\hat{\vec{x}}_i \approx \sum_j w_{ij} \vec{x}_j \qquad \text{for} \qquad \vec{x}_j \in N(\vec{x}_i) \tag{1}$$

that is, we may reconstruct a vector as a linear combination of its neighbors.

The LLE algorithm requires as inputs an $n \times m$ data matrix $X$, with rows $\vec{x}_i$, a certain number of dimensions $d < m$, and an integer $k > d+1$ to find local neighborhoods. The output is an $n \times d$ matrix $Y$, with rows $\vec{y}_i$. The LLE algorithm may be divided into three main steps (Roweis and Saul, 2000; Saul and Roweis, 2003):

Firstly, from each $\vec{x}_i \in R^m$, it finds the $k$ nearest neighbors. Subsequently, it finds the weight matrix $W$, which minimizes the reconstruction error for each data point $\vec{x}_i \in R^m$. Lastly, it finds the coordinates $Y$ which minimize the reconstruction error using the optimum weights. In the following section, we describe how to obtain the solution to each step of the LLE algorithm.

#### 2.1.1 Finding the local linear neighborhoods

A relevant aspect of the LLE is that this algorithm is capable of recovering embedding whose intrinsic dimensionality $d$ is smaller that the number of neighbors $k$. Moreover, the assumption of a linear patch enforces the existence of an upper bound on $k$. For instance, in highly curved datasets, it is not reasonable to have a large $k$, or that assumption would be then violated. In the uncommon situation where $k > m$, it has been demonstrated that each sample may be perfectly reconstructed from its neighbors. However, an additional problem then arises, namely the reconstruction weights are not unique anymore. To overcome such a limitation, some regularization is required in order to break the degeneracy (Saul and Roweis, 2003).

#### 2.1.2 Least-squares estimation of weights

The second step of the LLE algorithm consists of reconstructing each data point from its nearest neighbors. The optimal reconstruction weights may be computed in closed form. Without loss of generality, the local reconstruction error at point $\vec{x}_i$ may be expressed as follows:

$$E(\vec{w}) = \left\| \sum_j w_j (\vec{x}_i - \vec{x}_j) \right\|^2 \tag{2}$$

$$= \sum_j \sum_k w_j w_k (\vec{x}_i - \vec{x}_j)^T (\vec{x}_i - \vec{x}_k)$$

Defining the local covariance matrix $C$ as:

$$C_{jk} = (\vec{x}_i - \vec{x}_j)^T (\vec{x}_i - \vec{x}_k) \tag{3}$$

we find the following expression for the local reconstruction error:

$$E(\vec{w}) = \sum_j \sum_k w_j C_{jk} w_k = \vec{w}^T C \vec{w} \tag{4}$$

Regarding the constraint $\sum_j w_j = 1$, it may be understood in two different ways: geometrical and probabilistically. From a geometric point of view, it provides invariance under translation, adding a constant vector $\vec{c}$ to $\vec{x}_i$ and all of its neighbors while the reconstruction error remains unchanged. In terms of probability, enforcing the weights to sum to one results in $W$ becoming a stochastic transition matrix (Saul and Roweis, 2003). The estimation of the matrix $W$ reduces to $n$ eigenvalue problems. As there are no constraints across the rows of $W$, we may separately find the optimal weights for each sample $\vec{x}_i$, drastically simplifying the computations. Thus, there are $n$ independent constrained optimization problems expressed as follows:

$$\arg\min_{\vec{w}_i} \ \vec{w}_i^T C_i \vec{w}_i \quad \text{s.t.} \quad \vec{1}^T \vec{w}_i = 1 \tag{5}$$

for $i = 1, 2, ..., n$. Using Lagrange multipliers, we write the Lagrangian function as follows:

$$L(\vec{w}_i, \lambda) = \vec{w}_i^T C_i \vec{w}_i - \lambda(\vec{1}^T \vec{w}_i - 1) \tag{6}$$

Taking the derivatives with relation to $\vec{w}_i$:

$$\frac{\partial}{\partial \vec{w}_i} L(\vec{w}_i, \lambda) = 2C_i \vec{w}_i - \lambda \vec{1} = 0 \tag{7}$$

which leads to

$$C_i \vec{w}_i = \frac{\lambda}{2} \vec{1} \tag{8}$$

This is equivalent to solving the following linear system:

$$C_i \vec{w}_i = \vec{1} \tag{9}$$

and then normalizing the solution to guarantee that $\sum_j w_i(j) = 1$ by dividing each coefficient of the vector $\vec{w}_i$ by the sum of all the coefficients:

$$w_i(j) = \frac{w_i(j)}{\sum_j w_i(j)} \qquad \text{for} \qquad j = 1, 2, ..., m \tag{10}$$

If the number of neighbors $k$ is greater than the number of features $m$ then, in general, the space spanned by $k$ distinct vectors consists of the whole space. It means that $\vec{x}_i$ may be written precisely as a linear combination of its $k$-nearest neighbors. In fact, if $k > m$, then there are generally infinitely many solutions to $\vec{x}_i = \sum_j w_j \vec{x}_j$, because there would be more unknowns $k$ than equations $m$. In such a case, the optimization problem would be ill-posed, and regularization should be required. In all experiments described in the present paper, we set the regularization parameter as $\alpha = 0.001$.

### 2.1.3 Finding the coordinates

If the local neighborhoods are small enough compared to the curvature of the manifold, the optimal reconstruction weights in the embedding space as well as the weights reconstruction on the manifold are approximately the same. In fact, the two sets of weights are identical for linear subspaces, and for general manifolds they may be brought arbitrarily close to each other by shrinking the neighborhood. The key idea behind the third step of the LLE algorithm is to use the optimal reconstruction weights estimated by least-squares as the proper weights on the manifold and then solve for the local manifold coordinates. Thus, fixing the weight matrix $W$, the objective is to solve the following quadratic minimization problem:

$$\Phi(Y) = \sum_{i=1}^{n} \left\| \vec{y}_i - \sum_{j} w_{ij} \vec{y}_j \right\|^2 \tag{11}$$

In other words, we need to address the question about which coordinates $\vec{y}_i \in R^d$ (approximately on the manifold) are reconstructed by such weights $W$. In order to avoid degeneracy, we have to impose the two following constraints:

1. The mean of the data in the transformed space is zero, otherwise we would have an infinite number of solutions;

2. The covariance matrix of the transformed data is the identity matrix (i.e. there is no correlation between the components of $\vec{y} \in R^d$). This is a statistical constraint to assess that the output space is Euclidean, defined by an orthogonal basis.

However, differently from the estimation of the weights $W$, finding the coordinates does not simplify into $n$ independent problems, because each row of $Y$ appears in $\Phi$ multiple times, once as the central vector $y_i$ and also as one of the neighbors of other vectors. Thus, firstly, we rewrite equation equation 11 in a more meaningful manner using matrices:

$$\Phi(Y) = \sum_{i=1}^{n} \left( \vec{y}_i - \sum_{j} w_{ij} \vec{y}_j \right)^T \left( \vec{y}_i - \sum_{j} w_{ij} \vec{y}_j \right) \tag{12}$$

Applying the distributive law and expanding the summation, we then have:

$$\Phi(Y) = \sum_{i=1}^{n} \vec{y}_i^T \vec{y}_i - \sum_{i=1}^{n} \sum_{j} \vec{y}_i^T w_{ij} \vec{y}_j$$
$$- \sum_{i=1}^{n} \sum_{j} \vec{y}_j^T w_{ji} \vec{y}_i + \sum_{i=1}^{n} \sum_{j} \sum_{k} \vec{y}_j^T w_{ji} w_{ik} \vec{y}_k \tag{13}$$

Denoting by $Y$ the $d \times n$ matrix in which each column $\vec{y}_i$ for $i = 1, ..., n$ stores the coordinates of the $i$-th sample in the manifold and knowing that $\vec{w}_i(j) = 0$ unless $\vec{y}_j$ is one of the neighbors of $\vec{y}_i$, we may write $\Phi(Y)$ as follows:

$$\Phi(Y) = Tr(Y^T(I - W)^T(I - W)Y) \tag{14}$$

By defining the $n \times n$ matrix $M$ as follows:

$$M = (I - W)^T(I - W) \tag{15}$$

we get the following optimization problem:

$$\arg\min_{Y} \; Tr(Y^T M Y) \quad \text{subject to} \quad \frac{1}{n}Y^T Y = I \tag{16}$$

Therefore, the Lagrangian function is given by:

$$L(Y, \lambda) = Tr(Y^T M Y) - \lambda\left(\frac{1}{n}Y^T Y - I\right) \tag{17}$$

Lastly, differentiating and setting the result to zero leads to:

$$MY = \beta Y \tag{18}$$

where $\beta = \frac{\lambda}{n}$, showing that the $Y$ must be composed by the eigenvectors of the matrix $M$. As we have a minimization problem, we need to select the $d$ eigenvectors associated to the $d$ smallest eigenvalues to compose $Y$. Notice that $M$ consisting of an $n \times n$ matrix, it contains $n$ eigenvalues and $n$ orthogonal eigenvectors. Although the eigenvalues are real and non-negative, the smallest of them is always zero, with the constant eigenvector $\vec{1}$. This bottom eigenvector corresponds to the mean of $Y$ and should be discarded to enforce the constraint that $\sum_{i=1}^{n} \vec{y}_i = 0$ (de Ridder and Duin, 2002). Therefore, to get $\vec{y}_i \in R^d$, where $d < m$, we must select the $d + 1$ smallest eigenvectors and discard the constant eigenvector with zero eigenvalue. Specifically, we must select the $d$ eigenvectors associated to the bottom non-zero eigenvalues.

## 3 Parametric entropic LLE (PELLE)

The main motivation of the proposed parametric entropic LLE (PELLE) method is to find a surrogate for the local matrix $C_i$ for each sample of the dataset. Our method is a generalization of the regular LLE that deals with local Gaussian densities estimated along the patches of the KNN graph. It is worth noticing that, originally, $C_i(j, k)$ is computed as the inner product between $\vec{x}_i - \vec{x}_j$ and $\vec{x}_i - \vec{x}_k$, meaning that the we employ the Euclidean geometry in the estimation of the optimal reconstruction weights. In the proposed method, a non-linear distance function is adopted in the definition of such a matrix, being the relative entropy between Gaussian densities estimated within different patches of the KNN graph. Our inspiration is the parametric PCA method, an information-theoretic extension of the PCA method that uses the KL-divergence to compute a surrogate for the covariance matrix - i.e. entropic covariance matrix (Levada, 2020).

Let $X = \{\vec{x}_1, \vec{x}_2, \ldots, \vec{x}_n\}$, with $\vec{x}_i \in R^m$, be our data matrix. The first step of the proposed method consists of building the KNN graph from $X$. At this early stage, we employ the extrinsic Euclidean distance to compute the nearest neighbors of each sample $\vec{x}_i$. Denoting by $\eta_i$ the neighborhood system of $\vec{x}_i$, a patch $P_i$ is defined as the set $\{\vec{x}_i \cup \eta_i\}$. Notice that the number of elements of $P_i$ is $K + 1$, for $i = 1, 2, ..., n$, being a patch $P_i$ given by the following $m \times (k + 1)$ matrix:

$$P_i = \begin{bmatrix} x_i(1) & x_{i1}(1) & \dots & x_{ik}(1) \\ x_i(2) & x_{i1}(2) & \dots & x_{ik}(2) \\ \vdots & \vdots & \ddots & \vdots \\ \vdots & \vdots & \dots & \vdots \\ x_i(m) & x_{i1}(m) & \dots & x_{ik}(m) \end{bmatrix} \tag{19}$$

The idea behind the proposed method is to consider each column of the matrix $P_i$ as a sample of a multivariate Gaussian random variable of size $k+1$. Then, we compute the maximum likelihood estimators of the model parameters $\vec{\mu}_i$ (mean) and $\Sigma_i$ (covariance matrix) as follows:

$$\vec{\mu}_i = \frac{1}{k+1} \sum_{j=1}^{k+1} \vec{x}_{ij} \tag{20}$$

$$\Sigma_i = \frac{1}{k} \sum_{j=1}^{k+1} (\vec{x}_{ij} - \vec{\mu}_i)(\vec{x}_{ij} - \vec{\mu}_i)^T \tag{21}$$

Let $p(x)$ and $q(x)$ be multivariate Gaussian densities, $N(\vec{\mu}_1, \Sigma_1)$ and $N(\vec{\mu}_2, \Sigma_2)$. Then, the relative entropy $D_{\mathrm{KL}}(p\|q)$ becomes:

$$D_{\mathrm{KL}}(p\|q) = \frac{1}{2}\left[ log\left(\frac{|\Sigma_2|}{|\Sigma_1|}\right) + Tr\left[\Sigma_2^{-1}\Sigma_1\right] + (\vec{\mu}_2 - \vec{\mu}_1)^T \Sigma_2^{-1}(\vec{\mu}_2 - \vec{\mu}_1) - m \right] \tag{22}$$

As the relative entropy is not symmetric, it is then possible to compute its symmetrized counterpart as follows:

$$D_{KL}^{sym}(p\|q) = \frac{1}{2}\left[D_{KL}(p,q) + D_{KL}(q,p)\right] \tag{23}$$

which contains the following closed-form expression:

$$\begin{aligned} D_{KL}^{sym}(p\|q) = \frac{1}{2}\Bigg[ &\frac{1}{2}Tr\left(\Sigma_1^{-1}\Sigma_2 + \Sigma_2^{-1}\Sigma_1\right) \\ &+ \frac{1}{2}(\vec{\mu}_1 - \vec{\mu}_2)^T \Sigma_1^{-1}(\vec{\mu}_1 - \vec{\mu}_2) \\ &+ \frac{1}{2}(\vec{\mu}_2 - \vec{\mu}_1)^T \Sigma_2^{-1}(\vec{\mu}_2 - \vec{\mu}_1) - m \Bigg] \end{aligned} \tag{24}$$

In the proposed method, our goal is to approximate the multivariate normal density of the patch $P_i$ as:

$$p_i = \sum_j w_{ij} p_j \tag{25}$$

where $p_j \in N(p_i)$ denotes the multivariate normal densities from neighboring patches. Hence, we have to minimize the following quadratic error:

$$E(\vec{w}_i) = \left(p_i - \sum_j w_{ij} p_j\right)^2 = \left(\sum_j w_{ij}(p_i - p_j)\right)^2 \tag{26}$$

since the summation of the weights must be equal to one.

Defining the difference between two multivariate normal densities $p_i$ and $p_j$ as the symmetrized relative entropy, that is, $p_i - p_j = d_{ij} = D_{KL}(p_i \| p_j)$ we have:

$$E(\vec{w}_i) = \sum_j \sum_k w_{ij} d_{ij} d_{ik} w_{ik} \tag{27}$$

We compute the entropic matrix $C_i$ as follows:

$$C_i(j,k) = d_{ij} d_{ik} = D_{KL}(p_i \| p_j) D_{KL}(p_i \| p_k) \tag{28}$$

leading to the quadratic form:

$$E(\vec{w}_i) = \vec{w}_i C_i \vec{w}_i \tag{29}$$

Thus, we have the same optimization problem of regular LLE. To accelerate the computation of the proposed algorithm, we build a vector of relative entropies between $p_i$ and all the neighboring patches $p_j$, denoted by $\vec{d}_i$. The matrix $C_i$ is computed by the outer product of $\vec{d}_i$ with itself. It is worth noticing that distinctively from the standard LLE method, which employs the pairwise Euclidean distance, the proposed PELLE method employs a patch-based distance (i.e. relative entropy), becoming less sensitive to the presence of noise and outliers in the observed data. In the following section, we present computational experiments comparing the performance of the PELLE method against several popular manifold learning algorithms.

## 4 Fisher information and relative entropy

Let $p(X; \vec{\theta})$ be a probability density function, where $\vec{\theta} = (\theta_1, \ldots, \theta_k) \in \Theta$ is the vector of parameters. The Fisher information matrix is the natural Riemannian metric of the parametric space (Amari, 1985; 2000; Arwini and Dodson, 2008), being defined for $i, j = 1, \ldots, k$ as follows:

$$I(\vec{\theta})_{ij} = E\left[ \left( \frac{\partial}{\partial \theta_i} log\ p(X; \vec{\theta}) \right) \left( \frac{\partial}{\partial \theta_j} log\ p(X; \vec{\theta}) \right) \right] \tag{30}$$

The Fisher information matrix is the metric tensor that equips the underlying parametric space. It consists of the mathematical structure that defines the inner products in the local tangent spaces. The metric tensor enables the expression of the square of an infinitesimal displacement in the manifold $ds^2$ as a function of an infinitesimal displacement in the tangent space (Nielsen, 2020). In the case of a 2D manifold, it is given by a vector $[du, dv]$. Assuming a matrix notation we have:

$$ds^2 = \begin{bmatrix} du & dv \end{bmatrix} \begin{bmatrix} A & B \\ B & C \end{bmatrix} \begin{bmatrix} du \\ dv \end{bmatrix} = Adu^2 + 2Bdudv + Cdv^2 \tag{31}$$

where the matrix of coefficients $A$, $B$, e $C$ is the metric tensor. If the metric tensor is a positive definite matrix, the manifold is is known as Riemannian. It is worth noticing that in the Euclidean case, the metric tensor refers to the identity matrix - i.e., the space is flat, and we have the well-known Pythagorean relation $ds^2 = du^2 + dv^2$.

A relevant connection between the Fisher information and relative entropy is that the for nearby densities $p(x; \vec{\theta})$ and $p(x; \vec{\theta} + \Delta\vec{\theta})$, the KL-divergence becomes the Fisher information. Thus, it may be applied to approximate geodesic distances. The KL-divergence between two infinitesimally close densities may be expressed by a quadratic form, which coefficients are given by the elements of the Fisher information matrix. First, we recall that the symmetric KL-divergence between $p(x)$ and $q(x)$ is given by:

$$D_{KL}(p\|q) = \int (p(x) - q(x)) \, log\left(\frac{p(x)}{q(x)}\right) dx \tag{32}$$

Assuming we have a family of distributions parametrized by $\vec{\theta} = (\theta_1, ..., \theta_k)$ and $\vec{\theta'} = (\theta_1 + \Delta\theta_1, \theta_2 + \Delta\theta_2, ..., \theta_k + \Delta\theta_k)$ where $\Delta\theta_i$ is an infinitesimal displacement, then the symmetrized KL divergence is given by:

$$D_{KL}\left((p(x;\vec{\theta})\|p(x;\vec{\theta'})\right) =$$

$$= \frac{1}{2} \int \left[p(x;\vec{\theta}) - p(x;\vec{\theta'})\right] log\left(\frac{p(x;\vec{\theta})}{p(x;\vec{\theta'})}\right) dx \tag{33}$$

By definition, let the variation $\Delta p(x;\vec{\theta})$ be:

$$\Delta p(x;\vec{\theta}) = p(x;\vec{\theta}) - p(x;\vec{\theta'}) \tag{34}$$

which leads to:

$$D_{KL}\left((p(x;\vec{\theta})\|p(x;\vec{\theta'})\right) = \tag{35}$$

$$= \frac{1}{2} \int \frac{\Delta p(x;\vec{\theta})}{p(x;\vec{\theta})} log\left(\frac{p(x;\vec{\theta})}{p(x;\vec{\theta'})}\right) p(x;\vec{\theta}) dx \tag{36}$$

It is worth mentioning that the argument of the logarithm may be expressed as follows:

$$\frac{p(x;\vec{\theta})}{p(x;\vec{\theta'})} = \frac{p(x;\vec{\theta})}{p(x;\vec{\theta}) - \Delta p(x;\vec{\theta})} = \tag{37}$$

$$= \frac{p(x;\vec{\theta}) + \Delta p(x;\vec{\theta})}{p(x;\vec{\theta})} = 1 + \frac{\Delta p(x;\vec{\theta})}{p(x;\vec{\theta})}$$

Applying a Taylor approximation for the logarithm, for small values of $x$ we have:

$$log(1 + x) \approx x \tag{38}$$

Thus, we may write the following approximation:

$$log\left(1 + \frac{\Delta p(x;\vec{\theta})}{p(x;\vec{\theta})}\right) \approx \frac{\Delta p(x;\vec{\theta})}{p(x;\vec{\theta})} \tag{39}$$

which leads to:

$$D_{KL}\left((p(x;\vec{\theta})\|p(x;\vec{\theta'})\right) = \tag{40}$$

$$= \frac{1}{2} \int \left(\frac{\Delta p(x;\vec{\theta})}{p(x;\vec{\theta})}\right)^2 p(x;\vec{\theta}) dx$$

Considering that $\Delta p(x; \vec{\theta})$ is the arc length in the parametric space (manifold), we may express it in the tangent space as the following dot product:

$$\Delta p(x; \vec{\theta}) \approx \sum_{i=1}^{k} \left[ \frac{\partial}{\partial \theta_i} p(x; \vec{\theta}) \Delta \theta_i \right] \tag{41}$$

The above equation is the dot product between the gradient (local tangent coordinates), being $\Delta \vec{\theta}$ the displacement vector. Hence, we have:

$$\frac{\Delta p(x; \vec{\theta})}{p(x; \vec{\theta})} \approx \frac{1}{p(x; \vec{\theta})} \sum_{i=1}^{k} \frac{\partial}{\partial \theta_i} p(x; \vec{\theta}) \Delta \theta_i$$
$$= \sum_{i=1}^{k} \frac{\partial}{\partial \theta_i} log\ p(x; \vec{\theta}) \Delta \theta_i \tag{42}$$

Then, we may express equation equation 40 as:

$$D_{KL} \left( (p(x; \vec{\theta}) \| p(x; \vec{\theta}')) \right) = \frac{1}{2} \sum_{i=1}^{k} \sum_{j=1}^{k} g_{ij} \Delta \theta_i \Delta \theta_j \tag{43}$$

where

$$g_{ij} = \int \left( \frac{\partial}{\partial \theta_i} log\ p(x; \vec{\theta}) \right) \left( \frac{\partial}{\partial \theta_j} log\ p(x; \vec{\theta}) \right) p(x; \vec{\theta}) dx$$
$$= E \left[ \left( \frac{\partial}{\partial \theta_i} log\ p(x; \vec{\theta}) \right) \left( \frac{\partial}{\partial \theta_j} log\ p(x; \vec{\theta}) \right) \right] = I(\vec{\theta})_{ij} \tag{44}$$

In matrix vector notation, we have:

$$D_{KL} \left( (p(x; \vec{\theta}) \| p(x; \vec{\theta}')) \right) = \frac{1}{2} \Delta \vec{\theta}^T I(\vec{\theta}) \Delta \vec{\theta} \tag{45}$$

where $I(\vec{\theta})$ is the Fisher information matrix.

Therefore, adopting the symmetrized relative entropy as the similarity measure in PELLE means that we are approximating the geodesic distances between the multivariate Gaussian densities from neighboring patches along the KNN graph. As the variation between neighboring patches is smooth, the densities are similar.

## 5   Results

To test and evaluate the proposed method, we performed a set of experiments to compare the average classification accuracy obtained by four different supervised classifiers (KNN, decision trees, Bayesian classifier under Gaussian hypothesis, and random forest classifiers), after dimensionality reduction to 2D spaces. We then directly compare the proposed PELLE method against the PCA (Jolliffe, 2002), ISOMAP (Tenenbaum et al., 2000), LLE (Roweis and Saul, 2000), Hessian eigenmaps or HLLE (Donoho and Grimes, 2003), local tangent space alignment or LTSA (Zhang and Zha, 2004), and UMAP (McInnes et al., 2018) - which is considered as the state-of-the-art approach for manifold learning when the number of samples is large.

Subsequently to performing the dimensionality reduction-based metric learning for each dataset, we use 50% of the samples to train the supervised classifiers. Then, each one of them is used to classify the 50%

Table 1: Average classification accuracies produced by the PCA, ISOMAP, LLE, Hessian LLE, LTSA, UMAP, and PELLE method considering 25 data sets (2D case)

| Data set | PCA | LLE | ISOMAP | HLLE | LTSA | UMAP | PELLE |
|---|---|---|---|---|---|---|---|
| Bolts | 0.75 | 0.65 | 0.862 | 0.775 | 0.775 | 0.687 | **0.9** |
| Parity5 | 0.453 | 0.375 | 0.391 | 0.343 | 0.344 | 0.390 | **0.593** |
| Tic-tac-toe | 0.621 | 0.593 | 0.622 | 0.647 | 0.652 | 0.770 | **0.908** |
| Hayes-roth | 0.575 | 0.640 | 0.666 | 0.723 | 0.723 | 0.594 | **0.731** |
| Prnn_crabs | 0.605 | 0.655 | 0.600 | 0.640 | 0.637 | 0.747 | **0.867** |
| AIDS | 0.38 | 0.29 | 0.36 | 0.340 | 0.350 | 0.38 | **0.500** |
| Corral | 0.831 | 0.762 | 0.846 | 0.846 | 0.859 | 0.812 | **0.918** |
| Analcatdata_wildcat | 0.78 | 0.746 | 0.756 | 0.719 | 0.720 | 0.698 | **0.799** |
| Monks-problem-1 | 0.584 | 0.643 | 0.583 | 0.581 | 0.598 | 0.655 | **0.733** |
| Vineyard | 0.75 | 0.74 | 0.769 | 0.750 | 0.788 | 0.769 | **0.846** |
| Plasma_retinol | 0.534 | 0.553 | 0.541 | 0.533 | 0.534 | 0.530 | **0.603** |
| Visualizing_enviromental | 0.642 | 0.634 | 0.651 | 0.602 | 0.602 | 0.643 | **0.71** |
| Wine | 0.960 | 0.727 | 0.952 | 0.848 | 0.848 | 0.918 | **0.983** |
| Mu284 | 0.929 | 0.85 | 0.936 | 0.556 | 0.568 | 0.907 | **0.966** |
| Tae | 0.434 | 0.473 | 0.519 | 0.477 | 0.470 | 0.483 | **0.539** |
| Ar1 | 0.959 | 0.95 | 0.942 | 0.741 | 0.737 | 0.918 | **0.967** |
| Sa-heart | 0.651 | 0.658 | 0.683 | 0.634 | 0.633 | 0.663 | **0.712** |
| Kidney | 0.598 | 0.644 | 0.684 | 0.703 | **0.704** | 0.657 | **0.704** |
| Haberman | 0.736 | 0.692 | 0.745 | 0.722 | 0.723 | 0.707 | **0.753** |
| Lupus | 0.75 | 0.755 | 0.784 | **0.818** | **0.818** | 0.761 | 0.807 |
| Acute-inflammations | 0.933 | 0.8 | 0.954 | 0.783 | 0.770 | 0.945 | **0.991** |
| Chscase_geyser1 | 0.871 | 0.752 | 0.878 | 0.754 | 0.756 | 0.853 | **0.896** |
| Breast-tissue | 0.410 | 0.452 | 0.424 | 0.495 | 0.518 | 0.518 | **0.556** |
| Conference_attendence | 0.835 | 0.839 | 0.839 | 0.847 | 0.841 | 0.825 | **0.862** |
| Thoracic_surgery | 0.798 | 0.807 | 0.804 | 0.750 | 0.694 | 0.803 | **0.828** |
| **Average** | 0.695 | 0.667 | 0.712 | 0.665 | 0.666 | 0.705 | **0.787** |
| **Median** | 0.736 | 0.658 | 0.745 | 0.719 | 0.704 | 0.707 | **0.807** |
| **Minimum** | 0.380 | 0.290 | 0.360 | 0.340 | 0.344 | 0.380 | **0.500** |
| **Maximum** | 0.96 | 0.95 | 0.954 | 0.848 | 0.859 | 0.945 | **0.991** |

remaining samples from the test data set, being the average accuracy computed to evaluate the behavior of the dimensionality reduction in classification tasks. The results are reported in Table 1. In 24 out of 25 datasets, the proposed PELLE method obtains the highest classification accuracy, representing 96% of the cases. Despite being one of the best methods described in the literature, UMAP has a limitation: as it requires numerical optimization, it demands a sufficiently large number of samples for convergence to good results, making it suffer in datasets with limited number of samples. In this context, the proposed PELLE algorithm can be viewed as an interesting alternative for non-linear feature extraction in classification problems with a limited number of samples (small sample size problems).

To test whether the classification accuracy obtained through the proposed PELLE method are statistically superior to those obtained through the existing methods, we perform a non-parametric Friedman test (Friedman, 1937; Marozzi, 2014). Considering a significance level $\alpha = 0.01$, there is evidence to reject the null hypothesis that all groups are identical ($p = 1.36 \times 10^{-11}$). In order to check which groups are significantly different, we then performed a Nemenyi post-hoc test (Hollander et al., 2015). This test indicates that the PELLE method provides significantly higher classification accuracy levels compared to the PCA ($p < 10^{-3}$), ISOMAP ($p = 0.002$), LLE ($p < 10^{-3}$), Hessian LLE ($p < 10^{-3}$), LTSA ($p < 10^{-3}$), and UMAP ($p < 10^{-3}$).

Despite the superior results, especially in small sample size classification problems, it is worth mentioning that the proposed method has some caveats. A negative aspect of manifold learning algorithms, including

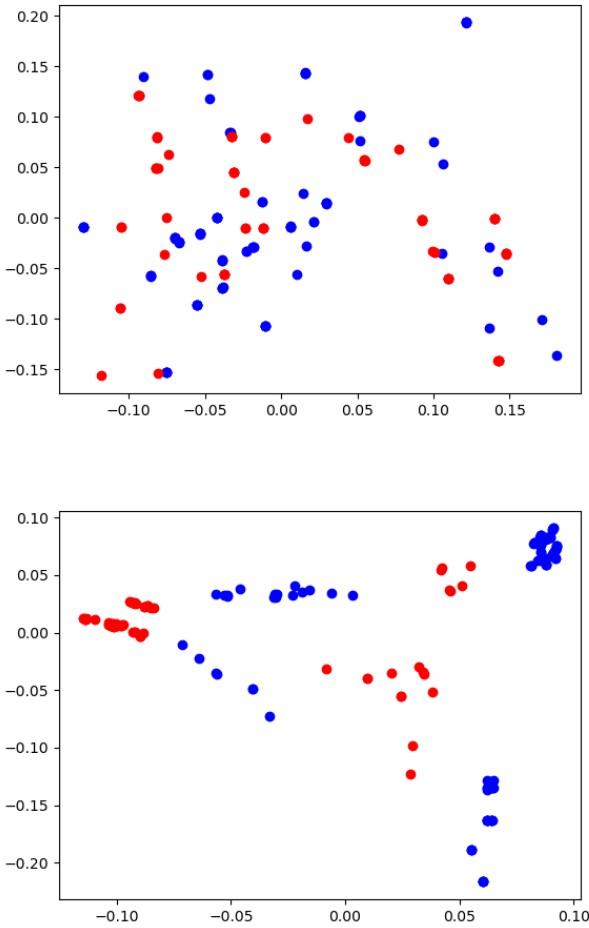

Figure 1: **Upper:** Data visualization comparing clusters obtained through the LLE. **Bottom:** PELLE method (K = 14) for the corral data set.

the proposed method, consists of the out-of-sample problem. Most unsupervised metric learning algorithms are not capable of dealing with new samples that are not part of the training data set in a straightforward manner. Another caveat of the proposed method concerns the definition of the parameter $K$ (i.e. number of neighbors) that controls the patch size. Our experiments reveal that the classification accuracy is sensitive to changes in such a parameter.

In summary, in the present study we employ the following strategy: for each data set, we build the KKN graphs for all values of K in the interval $[2, 40]$. We then select the best model as the one that maximizes the classification accuracy among all values of $K$. It is worth mentioning that we are using class labels to perform model selection, however the dimensionality reduction-based metric learning is fully unsupervised. A visual comparison of the clusters obtained through the LLE and PELLE method for the corral data set is depicted in Figure 1. It is worth noticing that the discrimination between the two classes is more evident through the results produced by the proposed PELLE method compared to the regular LLE, as there is less overlap between the blue and red clusters. An advantage of the proposed method is its capacity to learn good metrics even from small samples, while UMAP commonly requires more data to provide reasonably good results.

# 6 Conclusion

Unsupervised metric learning and manifold learning extract non-linear features from data, avoiding the Euclidean distance. Several manifold learning algorithms have been proposed in the pattern recognition literature, being the LLE method among the pioneering ones. Many LLE extensions have been devised to overcome limitations of the original method, such as the Hessian LLE (also known as Hessian eigenmaps), modified LLE, and LTSA. However, one yet unsolved problem refers to the fact that most LLE extensions use the Euclidean metric to measure the similarity between samples.

In the present paper, a parametric entropic LLE (PELLE) method is proposed to incorporate the relative entropy between local Gaussian distributions into the KNN adjacency graph. The rationale is to replace the pointwsie Euclidean distance by a patch-based information-theoretic distance to increase the robustness of the method against the presence of noise and outliers in the data. Our claim is that the proposed PELLE method is a promising alternative to the existing manifold learning algorithms, especially in small sample size classification problems. Our computational experiments support two main points. Firstly, the quality of the clusters produced by the PELLE method may be superior to those obtained by state-of-the-art manifold learning algorithms. Secondly, the proposed method features may be more discriminative in supervised classification than features obtained by state-of-the-art manifold learning algorithms.

Future works might explore the use of additional information-theoretic distances, such as the Bhattacharyya distance, Hellinger distance, and geodesic distances based in the Fisher information matrix. Another possibility is the non-parametric estimation of the local densities using kernel density estimation (KDE) techniques. In such a case, non-parametric versions of the information-theoretic distances might be employed to compute a distance function between the patches of the KNN graph. The $\epsilon$-neighborhood rule might also be used to build adjacency relations that define the discrete approximation for the manifold, leading to non-regular graphs. Furthermore, a supervised parametric entropic LLE might be created by removing the edges of the KNN graph for which the endpoints belong to different classes, enforcing the optimal reconstruction weights to use only the neighbors that belong to the same sample class.

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
