# OpenReview forum: "Parametric Entropic Locally Linear Embedding for Small Sample Size Classification"
_TMLR — Withdrawn by Authors_

### Review · Reviewer_uVn8 · 2024-05-23

**Summary Of Contributions:**

The paper proposed to replae the Eucliean distance in local linear embedding (LLE) with parametric entropic distances like the KL divergence, assuming multivariate Gaussian distribution of the local patches. Experiments on classification show that the proposed PELLE method outperforms other metric learning (dimensionality reduction) algorithms.

**Audience:**

Yes

**Claims And Evidence:**

Yes

**Requested Changes:**

Improve presentation, revise notations, adding more contents. For example, only classification accuracy was reported. Can you report more  metrics related to manifold preserving? Can you add some theoretical analysis for the approximation error?

**Strengths And Weaknesses:**

S1. The idea is to extend Euclidean distance based metric learning methods to parametric entropy based distances. While not particularly novel in terms of techniques, it is a good extension of existing algorithms.

S2. The experiments on 25 datasets seem promissing.

W1. The presentation can be improved. Please summarize the algorithm in an "Algorithm" environment. Also, what's the main conclusion of Section 4? Please summarize it in a theorem/proposition/claim.

W2. Some notations are confusing and not rigorous. For example, in Section 3, $p_j\in N(p_i)$ is used to denote the density funtions. But what is the definition of neighoring for densities? Later you wrote $p_i-p_j=d_{ij}=KL(p_i||p_j)$. This is not a rigorous notation. Please check all the notations.

W3. The idea seems too simple, so the content is not rich enough. A lot of space is used to introducing existing LLE algorithm. The proposed method only takes less than 2 pages. For a journal paper, it seems too short. It would be better if more contents can be provided, e.g., more theoretical analysis, experiments, etc.

W4. No theoretical guarantee on the approximation error.

---

### Review · Reviewer_Srpf · 2024-05-24

**Summary Of Contributions:**

The authors propose a new variant of the Locally Linear Embedding (LLE) method for "manifold-aware" dimensionality reduction - parametric entropic LLE, abbreviated as PELLE. The difference wrt LLE is in how weights w in the second step of the LLE algorithm are computed. Standard LLE relies on datapoints-pairwise distances within each kNN neighbourhood. PELLE fits a multivariate normal distribution to the datapoints in the neighbourhood of each datapoint, and then uses a symmetric KL between those distributions as a measure of distance. Authors motivate this design choice with the connection to the Fisher information matrix.
Authors show the proposed method outperforms other methods in this family on a classification task on several datasets.

**Audience:**

Yes

**Claims And Evidence:**

No

**Requested Changes:**

Please see the Weaknesses section above

**Strengths And Weaknesses:**

Disclaimer: review of low confidence (1 out of 5) - I am not familiar with the literature on the subject, and hence cannot comment on related work and novelty of the proposed method.

Strengths:
* S1. The proposed method does seem to outperform other methods according to the evaluation protocol adopted by the authors. (However I have doubts, questions, and requests regarding the experimental protocol)
* S2. Motivating the algorithmic change with the connection to Fisher information matrix is appealing, however I did not put in the effort to assess how accurate the portrayal of this relationship is.

Weaknesses:
* W1. Several claims unsupported (imo) by empirical evidence:
* W1A. “the proposed PELLE method employs a patch-based distance (i.e. relative entropy), becoming less sensitive to the presence of noise and outliers in the observed data.” - To my understanding, the sensitivity of the method to noise and outliers in the observed data is not explicitly assessed empirically or proved. Please provide empirical evidence or more thorough reasoning process to support this claim.
* W1B. “An advantage of the proposed method is its capacity to learn good metrics even from small samples, while UMAP commonly requires more data to provide reasonably good results.” - As above, please provide empirical evidence or more thorough reasoning process to support this claim. E.g. add the information about dimensionality & number of datapoints in each of the datasets evaluated on.
* W1C. “the quality of the clusters produced by the PELLE method may be superior to those obtained by state-of-the-art manifold learning algorithms” - Could you provide more reasoning/evidence than Figure 1, please?
* W1D. “one yet unsolved problem refers to the fact that most LLE extensions use the Euclidean metric to measure the similarity between samples.” - Could you spell out why is that a problem, please? The assumption made is that individual samples lie sufficiently close that we can consider them part of the same linear subspace.
* W2. Empirical evaluation
* W2A. Could you please add information about  the dimensionality & number of datapoints in each of the datasets evaluated on in Table 1?
* W2B. “We then select the best model as the one that maximizes the classification accuracy among all values of K.” Is this on the train set or on the test set? (since you don’t mention a validation dataset)
* W2C. I find it difficult to tell whether the classification performance is sufficient to characterize this method in comparison to other alternatives. Shouldn’t we also consider e.g. runtime/computational complexity?
* W2D. I would like to see the classification performance without averaging over the four supervised classifiers (KNN, decision trees, Bayesian classifier under Gaussian hypothesis, and random forest classifiers)
* W2E. Could you also add classification performance baselines of simply using each one of those four supervised classifiers without any dimensionality reduction, please? Without knowing those datasets I find it hard to know how much of a role dimensionality reduction plays here.
* W3. How tight the relationship between the Fisher information matrix and the D_KL distance metric is not entirely clear to me.
* W4. Writing is very generous. There are a lot of explicit derivations that are more suited for the appendices than the main text.
* W5. I would love for some of the ideas in the further work section (the last paragraph of the conclusion) to be included in this submission, as currently, especially given imo a rather brief experimental evidence section, feels not substantial enough for a journal publication.
* W6. typos
    * pointwsie
    * A, B e C
    * eq 45 orphaned opening parenthesis
    * “Subsequently to performing”
    * “express equation equation 40 as:”

---

### Review · Reviewer_xyZ1 · 2024-07-12

**Summary Of Contributions:**

This paper proposes a parametric entropy LLE (PELLE) method that incorporates the relative entropy between local Gaussian distributions in the KNN adjacency graph. Specifically, the proposed method replaces the pointwise Euclidean distance with a patch-based KL divergence to increase the robustness of the method against the presence of noise and outliers in the data.

**Audience:**

No

**Claims And Evidence:**

Yes

**Requested Changes:**

- The contribution of this paper seems to be mainly a method that replaces the Bhattacharyya distance of parametric PCA with the KL divergence and applies it to the prior stage of finding coordinates in LLE. This technical contribution is not sufficient and higher technical contributions are needed.
- Detailed ablation experiments and experiments with more realistic and larger data are needed.
- Many parts of the paper are devoted to content that has little relation to the essence of the proposed method. For example, the sections on finding coordinates in Section 2.1.3 and Fisher information and relative entropy in Chapter 4 have little relevance to the proposed method. These parts should be described more concisely.

- There are many typos or insufficient explanations in the paper's description. For example
1. In equation 1, \hat{x}=, not \hat{x}\approx
2. There is no definition of N(x_i) in Equation 1.
3. There is no precise definition of W, X or Y in page 2.
4. The assumption for the derivation of Equation 2 seems to require a constraint of \sum_j w_j =1, but there is no description of that constraint.
5. Equation 28 seems to be D_{KL}^{sym}, not D_{KL}.
6. In equation 29, w_i^T C_i w_i instead of w_i C-I w_i

**Strengths And Weaknesses:**

Pros.
- This paper proposes Parametric Entropic LLE, an improved method of locally linear embedding (LLE), in which each data point is reconstructed from its nearest neighbors in the second step of LLE. In this part, the proposed method applies Parametric PCA.
- The paper is simple and easy to understand.

Cons
- The paper claims to propose Parametric Entropic Locally Linear Embedding. The difference from ordinary LLE is that in the second stage of LLE, each data point is reconstructed from its nearest neighbors. The proposed method applies parametric PCA to this second part. However, the essence of LLE is the third step of finding the coordinates, but it seems that there is no special effort in this third step. This paper does not contribute to the core idea of LLE.
- The modification using Parametric PCA in the second stage of LLE also shows marginal novelty. The parametric PCA in the original paper uses the Bhattacharyya distance, while this paper proposes to use the KL divergence, which is different but essentially almost the same.
- In the derivation of equation 27, p_i-p_j = D_{KL}(p_i || p_j), which is a jump in logic. In addition, it seems to be D_{KL}^{sym}, not D_{KL}.

---

### Note · Authors · 2024-09-20

**Comment:**

We are working on the improvement suggestions made by the reviewers, which is taking longer than expected. In order to avoid impacting the (relatively short) review timing of this journal, we then decided to withdraw it. Maybe we should consider submitting it once more to the same journal after we finish this revision.

**Withdrawal Confirmation:**

I have read and agree with the venue's withdrawal policy on behalf of myself and my co-authors.